# Approaches for Inclusion Complexes of Ezetimibe with Cyclodextrins: Strategies for Solubility Enhancement and Interaction Analysis via Molecular Docking

**DOI:** 10.3390/ijms26041686

**Published:** 2025-02-16

**Authors:** Dae-Yeong Cho, Jeong-Gyun Lee, Moon-Jung Kim, Hyuk-Jun Cho, Jung-Hyun Cho, Kyeong-Soo Kim

**Affiliations:** 1Department of Pharmaceutical Engineering, Gyeongsang National University, 33 Dongjin-ro, Jinju 52725, Republic of Korea; daey2059@naver.com (D.-Y.C.); leepipi87@naver.com (J.-G.L.); kmjksk137@naver.com (M.-J.K.); 2Department of Innovative Drug Discovery and Development, College of Pharmacy, Keimyung University, 1095 Dalgubeoldaero, Dalseo-gu, Daegu 42601, Republic of Korea; hjcho89@kmu.ac.kr; 3Department of Pharmaceutical Engineering, Dankook University, 119 Dandae-ro, Dongnam-gu, Cheonan 31116, Republic of Korea

**Keywords:** ezetimibe, cyclodextrin, inclusion complexation, spray drying, solubility, dissolution, molecular docking

## Abstract

This study aimed to improve the solubility of ezetimibe (EZT), which has low aqueous solubility, by preparing complexes using β-cyclodextrin (β-CD) derivatives. Phase solubility studies and Job’s plot confirmed a high apparent stability constant for EZT with β-CD and even higher constants with its derivatives, establishing a 1:1 stoichiometric ratio. The composites were prepared using spray drying over a range of molar ratios, and their physicochemical properties were evaluated using techniques such as scanning electron microscopy (SEM), powder X-ray diffraction (PXRD), and Fourier transform infrared spectroscopy (FT-IR). Saturation solubility and in vitro dissolution tests revealed that solubility increased with higher CD molar ratios. EZT/RM-β-CD inclusion complexes (ICs) and EZT/DM-β-CD ICs exhibited a similar solubility, which was greater than that of EZT/HP-β-CD ICs and EZT/SBE-β-CD ICs (where RM, DM, HP, and SEB represent H, CH_3_, -CH_2_-CHOH-CH_3_ and -(CH_2_)_4_-SO_3_Na synthetic derivatives, respectively). Most complexes, except for EZT/SBE-β-CD at 1:2 or higher ratios, showed superior solubility compared with EZT powder and commercial products. Molecular docking simulations confirmed EZT inclusion within the CD, revealing hydrogen bonds and binding energies that aligned with solubility trends. These findings suggest that EZT complexes with β-CD derivatives significantly improve solubility, highlighting their potential for developing more effective oral solid formulations for hyperlipidemia treatment.

## 1. Introduction

Ezetimibe (EZT), chemically known as 1-(4-fluorophenyl)-3(R) [3-(4-fluorophenyl)-3(S)-hydroxypropyl]-4(S)-(4-hydroxyphenyl)-2 azetidinone (Figure 1A), is an active pharmaceutical ingredient (API) used to treat hypercholesterolemia by inhibiting the NPC1L1 protein, which is responsible for cholesterol transport [1,2,3]. This selective mechanism targets the intestinal absorption of plant sterols associated with biliary and dietary cholesterol, without affecting the absorption of fat-soluble vitamins, triglycerides, or bile acids [4,5]. Consequently, it effectively reduces elevated levels of total cholesterol, LDL-C, Apo B, and non-HDL-C in hyperlipidemic patients [6,7]. Despite these therapeutic benefits, EZT exhibits low solubility due to its high lipophilicity (log P = 4.56), resulting in low bioavailability [8,9]. However, due to good membrane permeability, it is classified as Class II in the Biopharmaceutics Classification System (BCS), affecting oral bioavailability [10,11,12]. To improve the solubility that affects the bioavailability of poorly soluble drugs like EZT, various solubilization techniques, such as co-crystallization [13,14], nanostructured lipid carriers (NLCs) [9,15], self-nanoemulsifying drug delivery systems (SNEDDSs) [16,17], nanosuspensions [18,19], and inclusion complexes [20] have been studied and continue to be actively researched. Among these, ICs using cyclodextrins (CDs) are widely utilized solubilization technologies, offering advantages such as improved aqueous solubility and physicochemical stability of drugs, the prevention of drug–drug interactions, minimized gastrointestinal and ocular irritation, and taste or odor masking [21]. CDs are cyclic oligosaccharides made of glucose units linked by α-1,4 glycosidic bonds, forming a toroidal shape with a hydrophilic exterior and hydrophobic core [22,23,24]. CDs can be classified into various types based on the number of glucopyranose units, of which α-CD, β-CD, and γ-CD contain six, seven, and eight glucopyranose units, respectively (Figure 1B) [25]. CDs exhibit a hydrophilic outer surface and a hydrophobic core, allowing for the formation of ICs where hydrophobic APIs can be hosted within the hydrophobic central cavity [26]. These guest/host ICs are primarily formed through non-covalent interactions, such as hydrophobic interactions, hydrogen bonding, and van der Waals forces [27]. This distinctive structure of CDs enhances the solubility, stability, and permeability of drug molecules, thereby increasing bioavailability through improved stability, solubility, wettability, and permeability [28,29]. Moreover, CDs can form complexes not only with drugs but also with contaminants and other target molecules, leveraging their ability to host a variety of compounds within the cavity [30,31]. CDs can also be chemically modified to improve their solubility and change the size of their internal cavities, and synthetic derivatives are classified into hydrophilic, hydrophobic, and ionizable groups. Among the derivatives of β-CD, for example, are hydrophilic hydroxypropyl-β-CD (HP-β-CD), hydrophobic heptakis(2,6-di-O-methyl)-β-CD (DM-β-CD), and ionizable sulfobutylether-β-CD (SBE-β-CD) (Figure 1C) [32,33]. Various methods, such as X-ray diffraction (XRD), Fourier transform infrared (FT-IR) spectroscopy, scanning electron microscopy (SEM), and solid-state nuclear magnetic resonance (ssNMR), are utilized to characterize ICs and to determine molecular interactions, surface morphology, and crystallinity [34,35,36]. Additionally, computer-based simulations, like molecular docking, have been crucial in drug discovery and understanding drug formulation interactions [37,38]. Among these, molecular docking is a simulation method used to predict the optimal structure of ligand–receptor complexes and evaluate the binding energy of these interactions [39,40,41]. EZT and CD ICs have previously been studied by preparing and comparing ICs using various methods such as co-evaporation and kneading with ternary ICs that involve hydrophilic auxiliary substances [42,43]. However, these studies did not provide explanations for the solubility differences between the ICs, or they only speculated. Additionally, studies analyzing the thermal characteristics of EZT and CD to confirm stability have been conducted [44]. However, these studies only prepared ICs using the co-evaporation method for β-CD and did not explain the solubility differences between the ICs, focusing instead on stability rather than evaluating the solubility. In contrast, we focused on enhancing solubility and prepared porous, solid dispersion-based ICs through spray drying for various β-CD derivatives and conducted solubility evaluations and molecular docking simulations. Through this approach, we provided structural insights into the ICs and considerations on the reasons for solubility differences.

This study aimed to prepare complexes using β-CD derivatives in order to improve the solubility of EZT. First, β-CD derivatives were selected through a CD screening process, with the physicochemical properties of the compounds used in this study summarized in Table 1. EZT and CD complexes were prepared using spray drying and the physicochemical properties of the prepared complexes were confirmed through XRD, FT-IR, and SEM analyses, with solubility being evaluated. Afterward, the optimal ICs were identified and visualized through molecular docking. Finally, we performed a discussion of the molecular docking results of the inclusion complexes. A comprehensive overview of the research design is shown in Figure 1.

## 2. Results and Discussion

### 2.1. Phase Solubility Studies and Job’s Plot

The phase solubility study of the drug was initially conducted in the presence of natural CDs, enabling the evaluation of the apparent stability constants (*K_s_*) and the selection of a natural CD for further derivative studies. Pure EZT has an intrinsic solubility (*S_o_*) of 0.43 µg/mL in water. In the phase solubility diagram, α-CD showed no increase in solubility with increasing concentrations, but EZT solubility showed a linear increase with increasing β-CD and γ-CD concentrations (Figure 2A). Among these, β-CD showed the highest solubility for EZT, with 9.31 ± 1.46 µg/mL at a concentration of 15 mM. Therefore, β-CD was selected, and additional phase solubility studies were conducted with its derivatives. The solubility of EZT was observed to increase linearly with increasing concentrations of RM-β-CD, DM-β-CD, TM-β-CD, HP-β-CD, and SBE-β-CD (Figure 2B). At a concentration of 15 mM, RM-β-CD and DM-β-CD exhibited similar solubilities of 42.89 ± 0.38 µg/mL and 45.35 ± 3.36 µg/mL, respectively, with DM-β-CD showing a slightly higher solubility. Subsequently, the compounds that showed high solubility, in order, were HP-β-CD (23.08 ± 0.33 µg/mL), SBE-β-CD (16.13 ± 0.18 µg/mL), and TM-β-CD (2.10 ± 0.02 µg/mL). According to the Higuchi–Connors classification, all of the β-CD derivatives in this experiment corresponded to A_L_-type curves with slopes of less than 1, suggesting that the stoichiometric ratio between the EZT and β-CD derivatives was 1:1. The slope, regression coefficient (*R*^2^), apparent stability constant (*K_s_*), and Gibbs free energy (Δ*G*) for each CD in the phase solubility plot are summarized in Table 2. *K_s_* was calculated using Equation (1) in Section 3.3. The calculated results show that DM-β-CD (7333.03 ± 417.29 M^−1^) > RM-β-CD (7023.74 ± 253.70 M^−1^) > HP-β-CD (3718.10 ± 318.19 M^−1^) > SBE-β-CD (2590.76 ± 258.70 M^−1^) > TM-β-CD (281.43 ± 68.88 M^−1^), which fall within the generally reported range for the *K_s_* of β-CD, 10~100,000 M^−1^ [47]. The higher the *K_s_*, the stronger the bond between the host and the guest (which indicates a good interaction between the drug and CD) and, thus, the more stable the complex that forms [48]. Therefore, the complex formed strongly and stably with EZT, as indicated by the value of the calculated stability constants above. Δ*G* was calculated using Equation (2) in Section 3.3. Both β-CD and its derivatives showed negative values, indicating that complex formation occurred spontaneously. Δ*G* had a larger negative value as the value of *K_s_* increased. This suggests that the larger the negative value of G, the higher the binding affinity between EZT and CD and the higher the stability of the complex [49]. However, among the β-CD derivatives, TM-β-CD showed a lower *K_s_* and a smaller negative value of Δ*G* for EZT than the parent β-CD. This is due to the unique cavity diameter of TM-β-CD, where the parent β-CD has diameters of 6.0 Å at the narrow rim and 6.5 Å at the wider rim, whereas TM-β-CD has diameters of 4.0 Å at the narrow rim and 7.0 Å at the opposite wider rim [50]. The narrow rim diameter of TM-β-CD is considered unsuitable for forming an IC with EZT, which likely accounts for the results observed above.

Job’s plot was constructed to further verify the stoichiometric ratio of the ICs alongside the phase solubility studies. The Δ*A* × *R* and *R* values for EZT and the CDs were calculated and used to create Job’s plot (Appendix A). In Job’s plot, the *R* value at the highest Δ*A* × *R* value indicates the stoichiometry of the complex. Specifically, an *R* value of 0.33 suggests a guest-to-host ratio of 1:2, an *R* value of 0.5 indicates a 1:1 ratio, and an *R* value of 0.66 corresponds to a 2:1 stoichiometry [51]. In this study, *R* = 0.5, which indicates that EZT formed a complex with all of the CDs used in the experiment at a 1:1 stoichiometric ratio (Figure 3). These findings are in agreement with the conclusions drawn from the phase solubility studies. Consequently, EZT/CD ICs were prepared in a 1:1 ratio.

### 2.2. Solubility Test of ICs

The saturated solubility of EZT powder and the prepared EZT/CD ICs were examined in distilled water (D.W.) and solutions with physiological pH conditions of 1.2, 4.0, and 6.8 [52]. This study was conducted to understand the solubility and pH-dependent solubility patterns, and the results for each solution are illustrated in Figure 4. The EZT powder exhibited very low solubility, approximately 0.4 μg/mL, in all solutions. However, the prepared EZT/CD ICs demonstrated a higher solubility than the EZT powder in all solutions, with a tendency for improved solubility as the CD molar ratio increased. Additionally, the EZT powder and the prepared EZT/CD ICs showed similar solubilities across different pH conditions, indicating that the solubility pattern is pH-independent, both for the EZT powder and its inclusion complexation with CD. In comparisons among the EZT/CD ICs, EZT/RM-β-CD and EZT/DM-β-CD exhibited a similar solubility, showing comparable solubility even at the same molar ratios. This result is attributed to their similar *K_s_*, leading to similar solubilization effects [53]. However, EZT/HP-β-CD and EZT/SBE-β-CD exhibited lower solubility, likely due to insufficient affinity resulting from lower *K_s_* values [54]. Therefore, it is expected that the HP-β-CD and SBE-β-CD used in previous studies will not be able to produce a great solubility improvement effect than RM-β-CD or DM-β-CD when preparing an IC with EZT. This can be seen as further progress in selecting the optimal IC that can maximize the solubility enhancement for EZT. Among the β-CD derivative ICs with EZT, EZT/SBE-β-CD showed the lowest solubility results and was only prepared and tested for solubility at molar ratios of 1/1 to 1/3. Higher-molar-ratio ICs were not prepared for EZT/SBE-β-CD due to it having a significantly lower solubility than the other EZT/CD ICs, leading to its exclusion.

### 2.3. Powder XRD Analysis (PXRD)

To confirm the formation of inclusion complexes (ICs) between EZT and CD, powder X-ray diffraction (PXRD) analysis was employed. The PXRD pattern of EZT exhibited several sharp diffraction peaks at the 2θ angles of 16.55°, 19.19°, 20.36°, 22.51°, 23.77°, 24.08°, 25.54°, and 29.88°, indicating that EZT is a crystalline drug. Conversely, the β-CD derivatives did not show any distinctive diffraction peaks, suggesting they are in an amorphous state. The physical mixtures (PMs) of EZT and CD exhibited PXRD patterns where the crystalline pattern of EZT was superimposed with the amorphous pattern of CD. However, in the PXRD diffraction patterns of the EZT/CD ICs, the crystalline pattern of EZT disappeared and an amorphous PXRD pattern was observed (Figure 5). This change indicates the formation of ICs due to interactions between EZT and CD, as well as the amorphization of the drug itself facilitated by the spray drying process [55,56].

### 2.4. Scanning Electron Micrographs

Scanning electron microscopy (SEM) was utilized to examine the microstructure and surface morphology of the ICs, SEM images were obtained. These images represent EZT, β-CD derivatives, EZT/CD ICs, and the PMs (Figure 6). EZT was observed to have an irregular polygonal shape (Figure 6A). The β-CD derivatives appeared as nearly spherical particles. The PMs were prepared by mixing EZT and CD in a 1:1 molar ratio in the solid state, with EZT present on the surface of the CD and no structural changes observed. This suggests that there is no inclusion interaction between EZT and β-CD derivatives in the solid state. Unlike the PMs, morphological changes in EZT and CD were observed in the EZT/CD ICs. The ICs with a 1:1 molar ratio all exhibited nearly spherical particles, whereas the ICs with a 1:6 molar ratio showed a morphology closer to fibers. This is believed to be because viscosity increases with the amount of polysaccharides, such as CD, resulting in a fiber-like form due to the high viscosity caused by excessive CD during spray drying [57]. All ICs showed alterations in drug particle form, signifying the emergence of a new solid phase [58]. Consequently, the morphological transformation of the EZT/CD ICs compared with EZT indicates interactions between EZT and CD [59].

### 2.5. Fourier Infrared Spectroscopy Analysis

To confirm the inclusion interactions between EZT and CD, FT-IR experiments were conducted (Figure 7). Analyses were performed on the EZT powder, β-CD derivatives, EZT/CD ICs, and PMs. In the EZT powder, strong absorption bands were observed at 3430 cm^−1^ and 3265 cm^−1^ (O-H stretching vibrations), 1728 cm^−1^ (C=O β-lactam stretching vibration), 1509 cm^−1^ (benzene ring C=C stretching vibration), 1399 cm^−1^ and 1224 cm^−1^ (C–F stretching vibration), and 832 cm^−1^ (para-substituted benzene ring vibration) [60,61]. In the PMs, except for the 1728 cm^−1^ and 1509 cm^−1^ peaks of EZT, the other peaks were not visible due to the peaks of CD. The spectrum appeared similar to a combination of EZT and CD without observable changes, suggesting no interaction [62]. However, noticeable changes in major peaks were observed in the EZT/CD ICs. In the EZT/DM-β-CD IC with a 1/6 molar ratio, the 1728 cm^−1^ peak of EZT shifted the most to 1745 cm^−1^, and the degree of the wavenumber shift increased with higher CD molar ratios. This indicates that the mentioned groups are directly involved in IC formation [63]. Additionally, it was observed in all EZT/CD ICs that the peaks at 1728 cm^−1^ and 1509 cm^−1^ decreased as the CD molar ratio increased. This suggests that EZT is included within the CD cavity due to interactions between these groups and CD, causing the EZT peaks to be obscured, and that greater inclusion occurs with higher CD molar ratios [64,65].

### 2.6. Dissolution Test

As EZT showed similar solubility in solutions under physiological conditions, an in vitro dissolution test was conducted for the EZT/CD IC in water over a period of 2 h (Figure 8). To assess dissolution behavior, comparisons were made with EZT powder and a commercial product (Ezetrol tablet). The low solubility of EZT powder rendered it undetectable, and the commercial product displayed limited solubility, with a dissolution endpoint of 0.42 ± 0.05 μg/mL. The EZT/CD IC exhibited a trend of increasing solubility with higher molar ratios of CD. For EZT/SBE-β-CD, the molar ratio of the EZT/SBE-β-CD IC was prepared and tested only up to 1/3, similar to the saturation solubility experiment. As a result, all IC ratios showed a lower solubility than the commercial product. However, among the other ICs, excluding SBE-β-CD, those with a molar ratio of 1/2 or higher showed a greater solubility than the commercial product, with higher initial dissolution rates and solubility observed as the molar ratio of CD increased. The EZT/RM-β-CD ICs and EZT/DM-β-CD ICs showed a similar solubility, with dissolution endpoint solubility at 4.92 ± 0.39 μg/mL for the EZT/RM-β-CD IC and 4.91 ± 0.11 μg/mL for the EZT/DM-β-CD IC at a 1/6 molar ratio. This was likely due to the similar degrees of substitution (DS) of the two β-CD derivatives (RM-β-CD DS: 12 and DM-β-CD DS: 14), resulting in similar solubilization effects [53]. In the IC with a 1/6 molar ratio, the EZT/HP-β-CD IC showed improved solubility with a dissolution endpoint of 3.83 ± 0.21 μg/mL, although it was lower than that of the methylated CDs. This could be attributed to the methylation of the CD ring by hydrophobic methyl groups, which expands the hydrophobic cavity and makes the interior more hydrophobic, resulting in higher solubility for the methylated CD EZT/CD ICs [66]. Overall, there was a trend of increasing solubility with higher CD molar ratios, similar to the solubility trend observed in the saturation solubility results. Except for the EZT/SBE-β-CD IC, the other EZT/CD ICs with a molar ratio of 1/2 or higher showed a greater solubility than the EZT powder and the commercial product, indicating that complexation with RM-β-CD, DM-β-CD, and HP-β-CD can enhance the low solubility of EZT, potentially enhancing oral absorption.

### 2.7. Molecular Docking

Molecular docking simulations were conducted to visualize and gain molecular insights into the potential ICs between EZT and CD. The visualized molecular docking structures with the lowest energy for each EZT/CD IC were obtained, confirming that EZT was included within the CD (Figure 9). In the case of the EZT/RM-β-CD IC, the oxygen atoms of the hydroxyl groups and the methylated methoxy groups of the CD formed hydrogen bonds with the polar hydrogens of EZT, with both polar hydrogens within the EZT molecule participating in hydrogen bonding. For the EZT/DM-β-CD IC, the polar hydrogen of EZT formed hydrogen bonds only with the oxygen atoms of the methylated methoxy groups of the CD, and, similarly to the EZT/RM-β-CD IC, both polar hydrogens within the EZT molecule were involved in hydrogen bonding. Unlike the previous two EZT/CD ICs, in the EZT/HP-β-CD IC and EZT/SBE-β-CD IC, only one hydrogen bond was observed between the polar hydrogen of EZT and the hydroxyl oxygen atom of the CD; additionally, the hydrogen bond lengths of these two EZT/CD ICs were longer, at 2.26 Å and 2.19 Å, respectively, than those in the previous EZT/CD ICs. This suggests that the ICs of methylated β-CD derivatives bind more strongly and stably than those of other β-CD derivatives. To compare the stability and interaction degree of the EZT/CD ICs, various energies such as van der Waals energy, hydrogen bonding energy, desolvation energy, electrostatic energy, torsional free energy, and the unbound system’s energy were calculated to obtain the final intermolecular energy and estimate the free energy of binding. The energy values and simulation rankings can be found in Appendix A, where the most negative energy results for each EZT/CD IC are also presented. A more negative final intermolecular energy indicates higher stability, and a more negative estimated free energy of binding reflects greater spontaneity in complex formation, indicating strong interactions between the drug and the CD. The correlation between these two types of energy and the *K_s_* values mentioned in Table 2 is shown in Figure 10. As a result, the linear equations were determined as y = −0.0004x − 5.957 for the estimated free energy of binding and y = −0.0004x − 8.344 for the final intermolecular energy. As lower energy values indicate stronger interactions, an increase in *K_s_* values corresponds to a decrease in energy values, demonstrating a negative correlation. Upon drawing trend lines for the data points, the R^2^ values were all above 0.77, indicating a strong correlation between the two types of energy and *K_s_*. Additionally, the trends in the two energy values for each EZT/CD IC were consistent with the those observed in the saturation solubility experiments and dissolution tests, serving as supporting data for the solubility differences among the EZT/CD ICs.

## 3. Materials and Methods

### 3.1. Materials

EZT was purchased from Ind-Swift Laboratories Ltd. (Chandigarh, India). Ezetrol^®^ tablets were supplied by Boryung Co. (Seoul, Republic of Korea). Hydroxypropyl-β-cyclodextrin (HP-β-CD), α-cyclodextrin (α-CD), β-cyclodextrin (β-CD), and γ-cyclodextrin (γ-CD) were supplied by Ashland Inc. (Wilmington, DE, USA). Randomly methylated cyclodextrin (RM-β-CD), heptakis(2,6-di-*O*-methyl)-β-cyclodextrin (DM-β-CD), and heptakis(2,3,6-tri-*O*-methyl)-β-cyclodextrin (TM-β-CD) were purchased from TCI Chemicals Co. (Tokyo, Japan). Sulfobutylether-β-cyclodextrin (SBE-β-CD) was purchased from GLPBIO Technology Inc. (Montclair, NJ, USA). Ammonium acetate, acetonitrile, and ethanol were purchased from Daejung Co., Ltd. (Siheung, Republic of Korea). Acetic acid was purchased from Samchun Chemical Co., Ltd. (Seoul, Republic of Korea). The deionized water used in the laboratory was produced using a distillation device. All other chemicals used were of analytical grade.

### 3.2. HPLC Conditions

The analysis of EZT was executed using an Agilent 1260 Infinity HPLC system (Agilent Technologies, Santa Clara, CA, USA) coupled with a UV–vis detector (Agilent G1314 1260, Agilent Technologies, CA, USA). The chromatographic separation employed a reversed-phase VDSpher 100 C18 M-E column (5 μm, 4.6 × 250 mm). The mobile phase consisted of a 100 mM ammonium acetate aqueous solution and acetonitrile, with the pH adjusted to 6.0 using acetic acid, in a 30:70 volume ratio. The flow rate was maintained at 1.0 mL/min, with an injection volume of 20 μL. UV detection was carried out at a wavelength of 232 nm, and data acquisition and processing were conducted using OpenLab CDS CS C.01.08 Chemstation software [67].

### 3.3. Phase Solubility Studies

Phase solubility studies were performed according to Higuchi and Connors [68]. An excess amount of EZT was added to each 1 mL aqueous solution containing increasing concentrations (0 to 15 mM) of α-CD, β-CD, γ-CD, RM-β-CD, DM-β-CD, TM-β-CD, HP-β-CD, and SBE-β-CD [69]. The mixtures were vortexed for about 2 min and then shaken at 75 rpm in a 37 °C water bath for 5 days [70]. After centrifugation at 13,500 rpm for 10 min, the supernatant was filtered through a 0.45 μm syringe filter to remove insoluble EZT. All samples were diluted with the mobile phase before being quantified using an HPLC system. A phase solubility diagram was obtained by graphing the concentration of EZT against that of CD. The apparent stability constants (*K_s_*) were calculated from the phase solubility diagram according to the following equation:(1)$$K_{S}=\frac{\left[D\;:CD\right]}{\left[D\right]\times [CD]}=\frac{Slope}{S_{O}(1-Slope)}$$
where *D* represents drug concentration and *S_o_* describes the solubility of EZT in the absence of CD [71].

The Gibbs free energy change (Δ*G*) during the inclusion process is a parameter that can be used to determine the stability and spontaneity of the complex [72]. It was calculated from the apparent stability constant (*K_S_*) using the following equation:(2)$$∆G=-RTlnK_{S}$$
where *R* is the universal gas constant (8.314 J/mol·K), and *T* is the experimental operating temperature (310 K). A greater negative Δ*G* value indicates that the IC formed between EZT and the CD is more stable [72,73,74].

### 3.4. Job’s Plot

The Job’s plot method of continuous variation is commonly used to determine the stoichiometry of the inclusion [65]. To perform the experiment, stock solutions with equimolar concentrations of CD and EZT were prepared. The total EZT and CD concentrations were kept constant (40 nM), and a series of EZT and CD mixtures were prepared, with the molar ratio of the two substances varying from 0.1 to 0.9 [75]. By plotting Δ*A* × *R* against *R*, Job’s plot was generated, where Δ*A* is the difference in the absorbance of EZT without and with CD, and *R* = [EZT]/([EZT] + [CD]) [76]. Δ*A* and *R* were measured using a UV–vis spectrophotometer (UV-1800, Shimadzu, Kyoto, Japan) at 232 nm.

### 3.5. Preparation of ICs

In the phase solubility study, β-CD was selected considering the EZT phase solubility results. Additionally, a phase solubility study was conducted for β-CD derivatives, selecting RM-β-CD, DM-β-CD, HP-β-CD, and SBE-β-CD, and ICs were prepared using the spray drying method. The ICs formed as compounds according to stoichiometry, prepared at the stoichiometric molar ratio between the host and guest. The EZT/CD ICs were prepared using a spray dryer (Yamato ADL311SA; Yamato Scientific Co., Ltd., Tokyo, Japan). EZT and CD were combined at various molar ratios (1/1~1/6) and dissolved in a mixed solvent of D.W. and ethanol via stirring. Once a homogeneously dissolved mixture was obtained, the mixed solution was spray dried. The spray drying conditions included an inlet temperature of 95 °C, an outlet temperature of 60–63 °C, a feed rate of 2.5 mL/min, and a spray air pressure of 0.1 MPa. The product yield of the obtained EZT/CD ICs was over 60% in all cases, and samples were stored in 70 mL HDPE containers with silica gel. The detailed composition of the spray drying solutions for the preparation of the EZT/CD ICs is shown in Table 3.

### 3.6. Solubility Test of ICs

Excess EZT/CD ICs were placed in 2 mL Eppendorf tubes. Then, 1 mL each of pH 1.2, pH 4.0, and pH 6.8 buffer solutions and D.W. were added to the tubes. Each sample was vortexed and shaken at 75 rpm and 37 °C in a water bath for 5 days. Afterward, the samples were centrifuged at 13,500 rpm for 10 min, and the supernatant was filtered through a 0.45 μm syringe filter to remove insoluble EZT. All samples were quantified using HPLC after dilution with the mobile phase. Each experiment was performed in triplicate [77].

### 3.7. Physicochemical Characterization of EZT/CD ICs

#### 3.7.1. Powder XRD Analysis

The EZT powder, EZT/CD PM (1/1 molar ratio), and EZT/CD IC samples were examined using a powder X-ray diffractometer (D/MAX-2500; Rigaku, Tokyo, Japan). A PXRD pattern analysis was conducted under the conditions of Cu-K*α* radiation (λ = 1.54178 Å) at a power setting of 40 kV and 100 mA. Additionally, an angular increase of 0.02° per second was selected to scan the 2θ angle range from 2° to 60° [78].

#### 3.7.2. Scanning Electron Micrographs

The surface characteristics of the EZT/CD powder and ICs were studied using a scanning electron microscope (Tescan-MIRA3; Tescan Korea, Seoul, Republic of Korea) under magnifications of ×1000 and ×10,000. All samples were fixed to stubs using double-sided adhesive tape. Then, all samples were coated with platinum using a sputter coater (K575X; EmiTech, Madrid, Spain) at a rate of 6 nm/min under vacuum (7 × 10^−3^ mbar) and placed under a scanning electron microscope to observe the surface morphological features [79,80].

#### 3.7.3. Fourier Infrared Spectroscopy Analysis

FT-IR spectra were measured on a Spectrum Two^TM^ (PerkinElmer, Waltham, MA, USA) spectrometer using the KBr plate technique. To prepare the KBr plate, 2 mg of solid IC and 200 mg of KBr were mixed. The IR spectrum was scanned 10 times at room temperature in the range of 4000 to 400 cm^−1^ [81].

### 3.8. Dissolution Test

In vitro dissolution tests of the EZT powder, Ezetrol tablets, and various molar ratios of the EZT/CD ICs were conducted using a USP dissolution apparatus II (RCZ-6N; Pharmao Industries Co., Shenyang, China). Each formulation, containing an equivalent of 10 mg of EZT, was placed into the dissolution tester. The dissolution test was performed at 37.0 ± 0.5 °C using 900 mL of D.W. as the dissolution medium, with the paddle speed set to 100 rpm. Samples were collected using a syringe, with 3 mL withdrawn at predetermined time intervals (5, 10, 15, 20, 30, 45, 60, 90, and 120 min). The collected samples were diluted two-fold with the mobile phase and filtered using a 0.45 µm syringe filter. The amount of EZT in the filtered samples was analyzed using the HPLC conditions described in Section 3.2. This dissolution test was performed six times for each formulation [82].

### 3.9. Molecular Docking

A docking study of EZT and CD was performed using AutoDock4 and AutoDock Vina implemented in AutoDockTools software, version 4.2.6 (MGL Tools version 1.5.7; Scripps Research Institute, La Jolla, CA, USA). The structures of RM-β-CD, DM-β-CD, HP-β-CD, and SBE-β-CD were reconstructed based on the coordinates of β-CD [83,84,85,86], with the initial structure of β-CD obtained from the protein complex structure (PDB: 1DMB). These structures were prepared using Avogadro 1.2.0 software and were subjected to geometrical optimization within the UFF force field (5000 steps, steepest descent algorithm) until no further energy minimization was observed [87]. The initial structure of EZT was downloaded as an SDF file format from PubChem and converted to a PDB file format using the OpenBabel-3.1.1 program [88], followed by geometrical optimization in the same manner as the cyclodextrins. Subsequently, both the host and guest were converted into PDBQT file format using AutoDockTools [89]. The optimized structures of EZT and the β-CD derivatives were utilized for molecular simulations. Non-polar hydrogen atoms were merged, and atomic charges were calculated using the Gasteiger–Marsili method [90]. Atomic types, hydrogen bond donors and acceptors, aliphatic and aromatic carbon atoms, and the rotatable bonds of the guest molecule were defined, with the guest EZT set as a flexible molecule and the host CD set as a rigid molecule. AutoGrid was used to calculate grid maps of interaction energies for the various atom types present in the docking guest. During docking, the grid map dimensions were set to 45 × 40 × 40 for the RM-β-CD system, 45 × 40 × 45 for the DM-β-CD system, 40 × 40 × 40 for the HP-β-CD system, and 110 × 90 × 90 for the SBE-β-CD system, with a grid point spacing of 0.375 Å to ensure that the CDs were completely surrounded [91]. To search for possible conformations of the EZT/CD ICs, the evaluation number and maximum generations of the Lamarckian genetic algorithm (LGA) were set to 2,500,000 and 27,000, respectively, with 100 docking calculations performed [92]. Finally, the complex with the lowest energy conformation, including free energy, in the docking simulation was obtained.

## 4. Conclusions

In this study, we aimed to improve the solubility of EZT by utilizing β-CD derivatives to prepare complexes. Phase solubility and Job’s plot evaluations revealed that EZT exhibited a high apparent stability constant with β-CD, and most of its derivatives showed even higher apparent stability constants. The stoichiometric ratio between EZT and CD was confirmed to be 1:1. Using spray drying, complexes were prepared at various molar ratios with the β-CD derivatives that demonstrated high apparent stability constants. The physicochemical properties of the prepared complexes were evaluated, followed by a saturation solubility assessment. As the molar ratio of CD increased, solubility tended to increase. The EZT/RM-β-CD IC and EZT/DM-β-CD IC exhibited a similar solubility, followed by the EZT/HP-β-CD IC and EZT/SBE-β-CD IC in a decreasing order of solubility. In vitro dissolution tests showed that, except for the EZT/SBE-β-CD IC, the complexes with molar ratios of 1:2 or higher had a greater solubility than the EZT powder and commercial products, with a similar trend observed in the saturation solubility assessment. Molecular docking simulations were performed to visualize and gain molecular insights into the prepared complexes. The results confirm that EZT was included within the CD, with hydrogen bonds forming between the two molecules. Furthermore, the binding energy values calculated through simulations showed a trend of large negative values, similar to those of the solubility assessments. Additionally, most of the previous studies evaluated complexes with HP-β-CD after preparation, contributing to the improvement of solubility. However, in this study, the discovery of methylated CD complexes that exhibit higher solubility than HP-β-CD and their evaluation through molecular docking have further enhanced the contribution to the research on ezetimibe inclusion complexes. In conclusion, the complexes formed between EZT and β-CD derivatives, with enhanced solubility, suggest the potential for development as oral solid forms for more effective hyperlipidemia treatment in the future.

## Data Availability

Data are available on request due to restrictions, e.g., privacy or ethical restrictions.

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
