# Peer review of "Approaches for Inclusion Complexes of Ezetimibe with Cyclodextrins: Strategies for Solubility Enhancement and Interaction Analysis via Molecular Docking"

_ijms, 2025, doi:10.3390/ijms26041686_

Round 1

Reviewer 1 Report

Comments and Suggestions for Authors

Thank you for submitting your manuscript titled“Approaches for Inclusion Complexes of Ezetimibe with Cyclodextrins: Strategies for Solubility Enhancement and Interaction Analysis via Molecular Docking” to the Journal of IJMS. I have completed my review and appreciate the effort and detail invested in your research, which addresses an important challenge: enhancing the solubility of ezetimibe (EZT) through complexation with β-cyclodextrin (β-CD) derivatives. The study's use of phase solubility analysis, physicochemical characterization, and molecular docking simulations provides valuable insights into the potential of these complexes for improved oral formulations of EZT.

However, I would like to highlight some important areas for improvement:

  1. Novelty and Contribution:
    The topic of EZT-cyclodextrin complexation has been previously explored in several studies, including:
    • Marta Biernacka, "Studies of the Formation and Stability of Ezetimibe-Cyclodextrin Inclusion Complexes."
    • Srivalli, Kale Mohana Raghava, and Brahmeshwar Mishra, "Improved aqueous solubility and antihypercholesterolemic activity of ezetimibe on formulating with hydroxypropyl-β-cyclodextrin and hydrophilic auxiliary substances," AAPS PharmSciTech 17 (2016): 272-283.
    • Patel, R., et al., "Solid-state characterization and dissolution properties of ezetimibe–cyclodextrins inclusion complexes," Journal of Inclusion Phenomena and Macrocyclic Chemistry 60 (2008): 241-251.

To strengthen the impact of your manuscript, please discuss these related works in the introduction, providing a comprehensive context for your research. Additionally, it is essential to clearly articulate the novelty of your study—specifically, how your experimental approach, findings, or implications differ from and advance upon the existing literature.

  1. Clarity in Results and Discussion:
    While the findings are promising, more explicit comparisons with previous studies could enhance the interpretation of your results. Highlighting unique aspects of your solubility enhancement strategies, stability constants, or docking simulations will further distinguish your work.

Addressing these points will significantly improve the clarity, relevance, and contribution of your manuscript. I look forward to reviewing a revised version of your work.

Sincerely,

Reviewer 2 Report

Comments and Suggestions for Authors

In this manuscript the Authors have presented the results of the study on the inclusion complexes formed between various CDs and ezetimibe, an important API. The complexes were synthesized and analyzed using various physicochemical methods.

The study is generally well prepared and described. I appreciate the careful analysis and discussion. However, the moderate similarity level (37%) indicates that some parts should be rephrased. However, in my opinion, the major drawback of this work is that the Authors haven’t mentioned that most of those complexes have been obtained and analyzed using the same methods before, i.e. the recently published work in the same journal (IJMS): 10.3390/ijms23010455 . The Authors don’t mention this work neither in the introduction nor in the discussion. The results of the current work should be carefully confronted with those presented in 10.3390/ijms23010455 . To be clear, this work ( 10.3390/ijms23010455 ) is not of mine nor my friends. Other comments and questions are presented below.

Line 45, what is directly impacting the bioavailability? I suppose it is solubility, but it should be stated.

Lines 56-58, this is not entirely true, as the large ring CDs exists and are being used as excipients

Lines 53, 59, those abbreviations (CD, API) should be defined earlier, the first time they appear in the text, precisely in lines 50 and 34, respectively.

Line 74, what about the application of Solid-State Nuclear Magnetic Resonance (ssNMR) for the analysis of CD-including systems? A suitable review should be cited as well.

Line 76, while those are indeed examples of the applications of in silico methods, it would be more reasonable to cite recent reviews on the application of both QM and MD simulations methods in the analysis of CD complexes.

Line 77, it should be “to predict the (optimal) structure”

Line 415, how many scans were recorded?

Line 432, how exactly the Authors have prepared the host molecules? More precisely, were all the hydroxyl groups substituted? If not, how the positions have been chosen?

Line 136, those CDs are usually called “native”, not “natural”

Figure 2, if it is not too much, I would kindly ask to use different colors for different CDs while presenting those results, especially in Figure 2B.

Table 2, the Ks values and their uncertainties are not rounded properly, the uncertainty should have either one or two significant digit(s).

2.3.Section, well, the method of choice in this case would be NMR as for the amorphous inclusion complexes PXRD is not very informative.

Figure 10, those values should be compared with those presented in Table 2. I would like to see the correlation (or its lack) and the discussion of such analysis.

Reviewer 3 Report

Comments and Suggestions for Authors

In the paper entitled “Approaches for Inclusion Complexes of Ezetimibe with Cyclodextrins: Strategies for Solubility Enhancement and Interaction Analysis via Molecular Docking”, the authors constructed several complexes of β-cyclodextrin (β-CD) and ezetimibe (EZT), the solubility of which in aqueous systems was low, for improving the solubility of EZT. The physicochemical properties of these complexes were evaluated by SEM, XRD, and FT-IR, and the results revealed that most of the complexes showed superior solubility to those of EZT and its commercial agents. This work was interesting and could provide a therectical basis for the development of new oral EZT formulations. Therefore, I recommend that this manuscript be accepted for publication in IJMS after minor revisions.

1)      What were the biological properties of these newly-prepared complexes? Compared with EZT and its commercial agents, these complexes exhibited more effectiveness in the treatment of hyperlipidemia or not?  

2)      In Figure 2A, several SD values might be too big. Please re-check the related results.

3)      There were still some errors in the English grammar, writing, and format. Moreover, according to the iThenticate report, the amount of wording duplication in the manuscript was 27%, which was too large. Please polish the manuscript.

Comments on the Quality of English Language

There were still some errors in the English grammar, writing, and format. Moreover, according to the iThenticate report, the amount of wording duplication in the manuscript was 27%, which was too large. Please polish the manuscript.

Round 2

Reviewer 1 Report

Comments and Suggestions for Authors

The manuscript presents findings related to the formation and stability of Ezetimibe-cyclodextrin inclusion complexes. However, a similar research topic has been previously explored in the study by Marta Biernacka, titled "Studies of the Formation and Stability of Ezetimibe-Cyclodextrin Inclusion Complexes." To strengthen the significance of your work, please clearly articulate the novel aspects that set your study apart from this prior research. Specifically, explain how your approach, findings, or applications contribute new insights beyond what has already been reported.

Additionally, related investigations have been presented in the following studies:
a. Srivalli, Kale Mohana Raghava, and Brahmeshwar Mishra. "Improved aqueous solubility and antihypercholesterolemic activity of ezetimibe on formulating with hydroxypropyl-β-cyclodextrin and hydrophilic auxiliary substances." AAPS PharmSciTech 17 (2016): 272-283.
b. Srivalli, Kale Mohana Raghava, and Brahmeshwar Mishra. "Improved aqueous solubility and antihypercholesterolemic activity of ezetimibe on formulating with hydroxypropyl-β-cyclodextrin and hydrophilic auxiliary substances." AAPS PharmSciTech 17 (2016): 272-283.
c. Patel, R., et al. "Solid-state characterization and dissolution properties of ezetimibe–cyclodextrins inclusion complexes." Journal of Inclusion Phenomena and Macrocyclic Chemistry 60 (2008): 241-251.

Citing these references in your introduction will provide a comprehensive background and contextualize your study within the existing body of research. Furthermore, elaborating on how your work advances or differentiates from these studies will enhance its impact and originality.

Reviewer 2 Report

Comments and Suggestions for Authors

The Authors have made some corrections, however, the work still requires major revisions. If the Authors refuse to apply those changes, or are unable to do them, I would suggest to reject this article.

I would presented the second review using style:

1.My original comment

2.Answer provided by the Authors

3.My new comment

However, in my opinion, the major drawback of this work is that the Authors haven’t mentioned that most of those complexes have been obtained and analyzed using the same methods before, i.e. the recently published work in the same journal (IJMS): 10.3390/ijms23010455 . The Authors don’t mention this work neither in the introduction nor in the discussion. The results of the current work should be carefully confronted with those presented in 10.3390/ijms23010455 . To be clear, this work (10.3390/ijms23010455 ) is not of mine nor my friends.

Although it may appear similar to the method mentioned in the referenced journal, this study involves the preparation of inclusion complexes using the spray-drying method. We aimed to produce and evaluate the optimal inclusion complexes with improved solubility by selecting β-Cyclodextrin derivatives through screening. Additionally, we performed molecular docking simulations to compare the inclusion complexes and provided an evaluation of the results. Furthermore, we identified inclusion complexes with superior solubility enhancement compared to those in the referenced journal, thereby increasing the contribution of our research.

I’m not stating that in your work there are no novel information. However, three out of 6 CDs used in this study has been previously utilized for the same purpose, to create the inclusion complexes with ezetimibe. The fact that you briefly mention that work solely in conclusions is significantly not enough. The results must be carefully compared and discussed, in order to analyze how the method of preparation affects the final results.

Line 74, what about the application of Solid-State Nuclear Magnetic Resonance (ssNMR) for the analysis of CD-including systems? A suitable review should be cited as well.

Since ssNMR analysis of the inclusion complex was not performed, it is not described separately, but we plan to attempt ssNMR analysis of the inclusion complex in the future.

The fact that a certain method has not been used in your study doesn’t mean that it should not be mentioned in the introduction, as ssNMR is the most informative method for analysis of the CD inclusion complexes in the solid state, certainly more important than PXRD

Line 76, while those are indeed examples of the applications of in silico methods, it would be more reasonable to cite recent reviews on the application of both QM and MD simulations methods in the analysis of CD complexes.

We know that applying QM and MD simulations can obtain more precise results. However, as with ssNMR, we do not cite or describe the papers that we applied because we plan to perform QM and MD simulations in the future.

What I try to say is that instead of citing the particular examples, it is more beneficial for the reader to cite recent review papers.

Line 432, how exactly the Authors have prepared the host molecules? More precisely, were all the hydroxyl groups substituted? If not, how the positions have been chosen?

Not all hydroxyl groups were replaced, and we checked the structures of each β-CD derivative in the references of the mentioned sentences and prepared them by working with Avogadro software.

This is not an answer to my questions, so I will repeat it: how the positions have been chosen?

Figure 10, those values should be compared with those presented in Table 2. I would like to see the correlation (or its lack) and the discussion of such analysis.

The following sentences have been added and modified to discuss the comparison of the final intermolecular energy and estimated binding free energy values in Figure 10 with the apparent stability constants and Gibbs free energy mentioned in Table 2 and the interpretation of similar results. Revised “Moreover, these two types of energies imply similar interpretations to Ks and G mentioned in Table 2.” in section “2.7 Molecular Docking”, page 12, line 304-305. Revised “This trend was also similar to Ks and G in Table 2. Furthermore, This trend is also consistent with the trends observed in the saturation solubility experiments and dissolution tests, and it can be used as supporting data for the solubility differences among the EZT/CD ICs.” In section “2.7 Molecular Docking”, page 12, line 308-311

This is not what I’ve aske the authors for, I would like to see the correlation (or its lack) and the discussion of such analysis. Please present another figure with the linear correlation, equation, R2 values, etc.

Round 3

Reviewer 1 Report

Comments and Suggestions for Authors

Accept in present form

Reviewer 2 Report

Comments and Suggestions for Authors

I'm very glad that finally the Authors have provided comprehensive answeres to my comments and questions. Current version can be accepted.